# The Secondary Structure of Potato Spindle Tuber Viroid Determines Its Infectivity in *Nicotiana benthamiana*

**DOI:** 10.3390/v15122307

**Published:** 2023-11-24

**Authors:** Yuxin Nie, Yuhong Zhang, Jian Wu

**Affiliations:** State Key Laboratory for Managing Biotic and Chemical Threats to the Quality and Safety of Agroproducts, Key Laboratory of Biotechnology in Plant Protection of MARA and Zhejiang Province, Institute of Plant Virology, Ningbo University, Ningbo 315211, China; 15169312489@163.com (Y.N.); yuhongzhang0105@139.com (Y.Z.)

**Keywords:** potato spindle tuber viroid, secondary structure, infectivity, mfold, *Nicotiana benthamiana*

## Abstract

The function of RNAs is determined by their structure. However, studying the relationship between RNA structure and function often requires altering RNA sequences to modify the structures, which leads to the neglect of the importance of RNA sequences themselves. In our research, we utilized potato spindle tuber viroid (PSTVd), a circular-form non-coding infectious RNA, as a model with which to investigate the role of a specific rod-like structure in RNA function. By generating linear RNA transcripts with different start sites, we established 12 PSTVd forms with different secondary structures while maintaining the same sequence. The RNA secondary structures were predicted using the mfold tool and validated through native PAGE gel electrophoresis after in vitro RNA folding. Analysis using plant infection assays revealed that the formation of a correct rod-like structure is crucial for the successful infection of PSTVd. Interestingly, the inability of PSTVd forms with non-rod-like structures to infect plants could be partially compensated by increasing the amount of linear viroid RNA transcripts, suggesting the existence of additional RNA secondary structures, such as the correct rod-like structure, alongside the dominant structure in the RNA inoculum of these forms. Our study demonstrates the critical role of RNA secondary structures in determining the function of infectious RNAs.

## 1. Introduction

In addition to carrying the instructions for protein synthesis, RNA is intricately involved in various aspects of gene expression and regulation due to its secondary structure [1,2,3]. The fundamental components of RNA secondary structure are characterized by distinct loops and simple helices [4]. Watson–Crick (WC) base pairs are the most prevalent type of canonical pairing found in RNA. Helical regions consist of consecutive WC base pairs, while loop regions are formed by a multitude of non-WC base pairs that are intricately arranged and terminate with one or more helices [5]. The characteristic functional structure of RNA is defined by the specific combinations of base pairings or conserved nucleotides within the loop and helical regions [6]. A variety of RNAs, including ribozymes, rRNAs, tRNAs, mRNAs, etc., exhibit catalytic or regulatory activities facilitated by their RNA secondary structure. Studies on translation have identified both tRNA abundance and mRNA secondary structure as significant factors influencing translation elongation rates [7,8]. Furthermore, the folded structure of rRNA might also contribute to translation catalytic processes [9]. Consequently, the intricacies of RNA folding serve as a critical bridge for comprehending the interplay between sequence and function. Site-directed mutagenesis has conventionally been employed to identify and investigate functional motifs within RNAs. In non-coding RNAs, even single-nucleotide variations (SNVs) can induce dynamic alterations in the RNA’s secondary structure, potentially resulting in functional impairment [10]. Nevertheless, certain mutational flexibility is granted through sequence covariation, enabling multiple nucleotide alterations while preserving the functional structure [11]. Hence, RNA sequences play a pivotal role in bridging the gap between structure and function.

In this study, potato spindle tuber viroid (PSTVd) was chosen as a model to achieve a more comprehensive understanding of the relationship between RNA secondary structure and function. The RNA genome of PSTVd comprises 359 nucleotides and adopts a rod-like structure characterized by 27 loops and 26 stems or helices [12,13,14]. After undergoing replication within the nucleus, facilitated by the host RNA polymerase II, PSTVd traverses between cells via plasmodesmata and can travel over long distances within the phloem. Specific loop and stem structures have been identified as key factors influencing the directional movement of PSTVd among various cell types [13,15,16,17,18,19]. Through the generation of linear RNA transcripts with distinct start sites (representing different PSTVd forms), we created a range of PSTVd secondary structures while retaining the identical sequence. Analysis of the PSTVd secondary structure using mfold revealed that the majority of the secondary structures exhibited a rod-like conformation. However, PSTVd forms beginning at positions 17, 250, and 265 displayed a distinctive ‘+’ or ‘Y’ shape. The predicted secondary structures were confirmed by native PAGE gel electrophoresis. Plant infection assays have revealed that the formation of a proper rod-like structure is crucial for the successful infection of PSTVd. Interestingly, the inability of PSTVd forms with non-rod-like structures to infect plants could be partially compensated by increasing the amount of linear RNA inoculum. These studies confirm the intrinsic significance of RNA sequences in bridging structure and function. They also underscore that the establishment of a correct rod-like structure is essential for successful infection. Our findings will contribute to a deeper understanding of the relationship between RNA structure and function.

## 2. Materials and Methods

### 2.1. Secondary Structure Prediction, Construction of 12 PSTVd Forms, and In Vitro Transcription

Mfold (http://www.unafold.org/mfold/applications/rna-folding-form.php, accessed on 1 March 2023) was employed to predict the secondary structure of the linear form of PSTVd intermediate form with different starting sites, based on the concept of minimal free energy [20]. Only the one with the minimal free energy at 37 °C was used.

In vitro transcripts of the PSTVd intermediate form were generated using plasmid pRZ6-2 as a template. Plasmid pRZ6-2, which includes a cDNA copy of the PSTVd sequence adjacent to a T7 promoter sequence, was generously provided by Dr. Robert Owens. The circular form of PSTVd intermediate strain, denoted as PSTVd-C and characterized by its rod-like structure, was synthesized following a previously described procedure [17]. Using PSTVd-C as a template, RT-PCR was employed with primers containing the T7 promoter region to generate a total of 12 PSTVd forms: PSTVd-2, -16, -17, -36, -86, -132, -174, -197, -250, -265, -319, and -326. These forms were designated based on the positions of their respective start sites. For example, a PSTVd form originating from position 2 was labeled as PSTVd-2. Primers used in RT-PCR were listed in Table 1. The amplified products are 359 nucleotides in size, matching the genome size of the PSTVd intermediate strain. RT-PCR products were purified through gel purification and subjected to in vitro transcripts using the T7 MEGAscript kit (ThermoFisher Scientific, Waltham, MA, USA) to prepare RNA transcripts for rub-inoculation.

### 2.2. Plant Materials and Inoculation

*N. benthamiana* plants were cultivated in a greenhouse with supplementary lighting. The temperature was kept at 23 °C with a photoperiod of 16 h of light followed by 8 h of darkness. As mentioned above, in vitro transcripts of the PSTVd intermediate strain were generated through in vitro transcription using plasmid pRZ6-2 as a template. In vitro transcription was conducted using the MEGAscript™ T7 Transcription Kit at 37 °C overnight (Cat # AM1334, Thermo Fisher Scientific, Shanghai, China). Rub-inoculation was conducted on 2-week-old plants possessing two fully expanded leaves. In essence, these two leaves were gently dusted with carborundum using a brush, and approximately 150 or 1500 ng of RNA in vitro transcript was introduced onto each leaf, resulting in a cumulative amount of 300 or 3000 ng of RNA for each plant. DEPC water inoculation was included as a mock experiment.

### 2.3. PAGE Gel Electrophoresis

The in vitro transcripts were initially purified by undergoing gel purification on a 5% urea-PAGE gel (200 V at room temperature for 1 h). Following this, in vitro RNA folding was performed according to previously described methods [19]. Briefly, the in vitro transcripts were first dissolved in 0.5× TE buffer at pH 8.0. They were then subjected to heating at 95 °C for 3 min, followed by cooling on ice for 5 min. Subsequently, the RNA was incubated at 37 °C for 5 min in a folding buffer consisting of 500 mM Tris–HCl at pH 7.5 and 500 mM NaCl. After adding MgCl_2_ to a final concentration of 10 mM, the mixture was further incubated for 30 min. Electrophoresis was then performed using 5% native PAGE gel (80 V for 4 h at 4 °C), with each sample containing 50 ng of RNA. For comparison, PSTVd-C was included as a control [21]. Trans2K^®^ Plus II DNA Marker (Transgen Biotech, Peking, China) was also loaded as a size marker. Images were captured after ethidium bromide staining under UV light.

### 2.4. RNA Extraction and RNA Blot

Leaf samples were collected from the top two fully expanded leaves at 28 days post-inoculation (dpi). Total RNA was extracted using Trizol reagent according to the manufacturer’s instructions (Invitrogen, Carlsbad, CA, USA). RNA blot was performed as previously described [16]. Roughly 10 mg of RNA samples were loaded onto a 5% polyacrylamide gel containing 8 M urea followed by electrophoresis using Bio-Rad Mini-PROTEAN Tetra Cell. These samples were then transferred onto a Hybond-XL nylon membrane (Amersham Biosciences, Piscataway, NJ, USA) using a Trans-Blot^®^ SD Semi-Dry Transfer Cell (Bio-Rad, Shanghai, China) and immobilized through UV-cross-linking. Hybridization was conducted at 65 °C using DIG-11-UTP-labeled riboprobes. These riboprobes were prepared through in vitro transcription using the T7 Maxiscript kit (ThermoFisher Scientific), with the *SpeI*-linearized pInter(–) plasmid serving as the template [19]. Following an overnight hybridization, the membranes were subjected to two rounds of washing at 65 °C, first in a solution of 2× SSC (where 1× SSC contains 0.15 M NaCl and 0.015 M sodium citrate) with 0.1% sodium dodecyl sulfate (SDS) and then in 0.2× SSC-0.1% SDS. Signal detection was accomplished using the MYECL™ Imager (Thermo Fisher Scientific, Shanghai, China).

## 3. Results

### 3.1. Construction of PSTVd Forms and Secondary Structure Prediction

The PSTVd genome consists of 359 nucleotides. Predictions were made for the secondary structure of forms starting at each of the 359 positions. However, only twelve positions were chosen to effectively represent various regions of the genome and different types of secondary structures. Using the PSTVd intermediate strain as a template, linear RNA transcripts with varying start sites were generated, resulting in distinct PSTVd forms. These forms were named based on their specific starting positions. For instance, the PSTVd form beginning at position 2 was assigned as PSTVd-2. The twelve PSTVd forms were PSTVd-2, -16, -17, -36, -86, -132, -174, -197, -250, -265, -319, and -326 (Figure 1A). The secondary structure of the PSTVd intermediate strain resembles a rod-like shape, the five common domains found in members of the *Pospiviroidae* family were presented (Figure 1A) [12,13,14,15]. The starting sites of some of the 12 forms are mapped to the two conserved regions. Specifically, the starting sites of PSTVd-16 and PSTVd-17 were located on the TCR, while those of PSTVd-86, PSTVd-250, and PSTVd-265 were found in the CCR. The secondary structures of these 12 PSTVd forms were predicted using mfold (http://www.unafold.org/mfold/applications/rna-folding-form.php, accessed on 1 March 2023) [20]. Among these 12 forms, nine were predicted to form a rod-like structure. PSTVd-17 and PSTVd-250 formed a ‘Y’ shaped structure, while PSTVd-265 adopted a ‘+’ shaped structure (Figure 1B).

### 3.2. Secondary Structure Verification through Native PAGE Gel Electrophoresis

Electrophoresis enables the evaluation of RNA based on its size and quantity. Nevertheless, in native gels, the electrophoretic mobility of RNA is influenced not only by its size but also by its secondary structure. In this study, all 12 PSTVd forms share identical sequences and sizes. Consequently, the electrophoretic mobility of these 12 forms is exclusively influenced by their respective secondary structures within the context of native PAGE gel. To examine the secondary structure of the 12 PSTVd forms through electrophoresis, full-length PSTVd RNA transcripts were generated via in vitro transcription. This process utilized PCR products comprising the entire PSTVd genome, initiated from specific sites along with the T7 promoter, as the template. The in vitro transcripts were initially purified via gel purification using a 5% urea-PAGE gel, followed by their separation through 5% native PAGE gel electrophoresis (80 V for 4 h at 4 °C), with each sample containing 50 ng of RNA. The circular form of PSTVd (PSTVd-C), characterized by a rod-like structure, was synthesized using a procedure previously described and included as a control [21]. Three biological replicates (reps) were included. It was observed that all forms, except for PSTVd-17, PSTVd-250, and PSTVd-265, exhibited similar electrophoretic mobility in comparison to PSTVd-C (Figure 2). However, PSTVd-17, PSTVd-250, and PSTVd-265 displayed significantly slower movement, indicating that these three forms adopt distinct secondary structures distinct from the rod-like shape. Therefore, the predicted secondary structures were confirmed.

### 3.3. Secondary Structure Determines the Infectivity of PSTVd

It is well-established that RNA’s secondary structure plays a crucial role in determining its function. Nevertheless, conventional investigations into the interplay between RNA secondary structure and function often necessitate the application of mutagenesis, leading to sequence alterations. Consequently, the potential impact of RNA sequence on RNA function cannot be disregarded. In this study, a collection of 12 PSTVd forms sharing identical sequences was established. These forms were categorized into three groups based on their secondary structures: a rod-like shape group consisting of PSTVd-2, -16, -36, -86, -132, -174, -197, -319, and -326; a ‘Y’ shape group comprising PSTVd-17 and -250; and a ‘+’ shape group represented by PSTVd-265. These three groups of forms underwent a plant infection assay using *N. benthamiana* plants, with six plants assigned to each form. The inoculation was carried out through rub-inoculation, 300 ng PSTVd in vitro transcripts per plant. Systemic infection was checked at 28 dpi through RNA blot. It was observed that all forms in the rod-like shape group infected 6/6 plants, while PSTVd-17 in the ‘Y’ shape group failed to infect any plant, and another form PSTVd-250 in this group only infected 2/6 plants (Figure 3, Table 2). The only form PSTVd-265 in ‘+’ shape group infected 0/6 plant. Our data strongly suggest that the formation of a rod-like structure is a prerequisite for the successful infection of PSTVd.

A mock (M) experiment involved DEPC water inoculation, while the positive (P) control comprised a PSTVd-positive RNA sample derived from a PSTVd-infected *N. benthamiana* plant. The position of circular PSTVd is marked by PSTVd-C. Ethidium bromide staining of rRNA was utilized as a loading control.

### 3.4. Enhancing the RNA Inoculum Quantity Partially Compensated for the Infectivity of PSTVd Forms Exhibiting Non-Rod-Like Structures

It is known that RNA structures are dynamic and often resist forming uniform configurations [22]. PSTVd forms in the ‘Y’ shape and ‘+’ shape groups may predominantly adopt non-rod-like structures; nevertheless, this does not preclude the potential for a small subset of RNA molecules to correctly form rod-like structures. Therefore, increasing the amount of RNA inoculum may compensate for the infectivity of PSTVd forms exhibiting non-rod-like structures. To explore this possibility, the plant infection assay was repeated using the PSTVd-17, PSTVd-250, and PSTVd-265 forms, employing an RNA inoculum of 3000 ng for each plant. The assessment of systemic infection took place at 28 dpi, revealing that PSTVd-17, PSTVd-250, and PSTVd-265 affected 2/6, 4/6, and 2/6 plants, respectively. In comparison to the initial infection assay which utilized 300 ng of RNA inoculum for each plant, notable increases in infection rates were observed across all three forms (Figure 4, Table 2). These findings strongly suggest that increasing the quantity of RNA inoculum partially compensated for the reduced infectivity observed in PSTVd forms exhibiting non-rod-like structures. Furthermore, these results also imply the potential existence of a minor subgroup of RNA molecules within these three forms that can adopt the correct rod-like structure.

## 4. Discussion

In this study, the relationship between RNA secondary structures and biological function was comprehensively explored using PSTVd, a highly structured and self-replicating infectious RNA, as the model system. We showed that forming a rod-like structure is a prerequisite for the successful infection of PSTVd. Additionally, even for PSTVd forms predicted to form non-rod-like structures, a minor group of RNA molecules may still form the correct rod-like structure, and increasing the quantity of RNA inoculum may partially compensate for the infectivity.

Viroids represent a class of infectious agents consisting of single-stranded circular RNA molecules, typically measuring around 250–400 nucleotides in length [23]. These covalently closed circular RNA infectious molecules, although remarkably simple in structure, have the capacity to induce infections in a wide range of higher plants [24]. This can result in the development of diseases that have the potential to cause substantial agricultural harm [25]. Previous studies on the secondary structure of viroids have shown that *Pospiviroidae* family members (such as PSTVd) assume a secondary structure resembling a rod, whereas viroids from the *Avsunviroidae* family exhibit a native state secondary structure that is either branched or quasi-rod-like [26,27,28]. The functionality of two viroid families is intricately connected to their specific secondary structures. Among the *Pospiviroidae* family members, the secondary structure displays five distinct domains that hold significance in terms of both structure and function: the C domain, the pathogenicity domain, the variable domain, as well as two terminal domains. Particularly noteworthy is the CCR situated within the C domain, which serves as a defining trait of *Pospiviroidae* viroids. The particular secondary structures in this region play a crucial role in governing viroid replication and processing. In contrast, viroids belonging to the *Avsunviroidae* family lack a CCR. However, they do exhibit a self-cleaving ability facilitated by the formation of hammerhead ribozyme structures [26,29]. Moreover, during replication and processing, viroids can adopt alternative structures. Examples include the formation of extra-stable hairpins in *Pospiviroidae* members and self-splicing conformations in *Avsunviroidae*, facilitating the progression of the life cycle [30,31]. Because there is no protein coat encasing the viroid genome, the viroid’s secondary structure assumes a pivotal role in dictating its capacity to invade plant cells, ensure its survival, and induce pathogenic effects [14]. Unlike other studies that focus on the relationship between RNA secondary structure and function through RNA sequence modifications, we created linear RNA transcripts with different start sites. This approach enabled us to establish a total of 12 PSTVd forms with distinct secondary structures while retaining the exact same sequence. In the plant infection assay, only those with a proper rod-like structure were able to successfully infect all the plants. Our thorough analysis validated the significance of the rod-like structure in PSTVd infection. It is unlikely that the altered infectivity of three PSTVd forms, namely, PSTVd-17, PSTVd-250, and PSTVd-265, is solely attributed to the presence of unclosed nicks in their functional regions. For instance, both PSTVd-16 and PSTVd-17 were mapped to the TCR region, yet they exhibited different infectivities. Similarly, PSTVd-86, PSTVd-250, and PSTVd-265 were all mapped to the CCR region, but they also displayed varying infectivities.

Interestingly, by increasing the RNA inoculum amount, the infection rate of three PSTVd forms that were initially predicted to adopt non-rod-like structures was effectively enhanced. It has been noticed that several RNAs possess alternative structures that correspond to different functions. For instance, the 5′ end of *Leptomonas collosoma* spliced leader RNA can interchange between two alternative structural forms to affect RNA splicing [32]. In effect, it has been demonstrated that the formation of viroid RNA structure heavily depends on the dynamic competition between alternative RNA structures [33]. We entertained the idea that even within the three forms anticipated to adopt ‘Y’ or ‘+’ shaped structures, a minority of RNA molecules might assume the appropriate rod-like conformation necessary for plant infection. Additionally, augmenting the RNA inoculum could potentially boost infection rates by increasing the quantity of these particular RNA molecules.

In this study, we utilized three forms exhibiting non-rod-like structures and nine forms with rod-like structures. While previous studies have explored the infectivity of various inocula, such as monomeric transcripts, minus strand oligomeric PSTVd, and transcripts with different starting positions [34,35,36,37], our study is the first to specifically focus on the secondary structure. Intriguingly, reduced infectivity was exclusively observed in the three forms with non-rod-like structures, while the nine forms with rod-like structures remained unaffected. Increasing the amount of RNA transcripts led to an enhanced infection rate in all three non-rod-like forms. However, it is important to note that our study employed only six plants for each form. Therefore, our findings may warrant further validation through future studies involving a larger number of plants for each form.

## 5. Conclusions

In conclusion, our study employed PSTVd as a model to investigate the connection between RNA structure and function. By generating a total of 12 PSTVd forms with the same sequence but different structures, we highlighted the critical role of a specific rod-like structure for successful infection. Our findings underscore the importance of accurate secondary structures in influencing the functionality of infectious RNAs, thereby enhancing our comprehension of their intricate mechanisms. Our study also provides guidance for future studies on the design and preparation of infectious clones of viruses and viroids.

## Figures and Tables

**Figure 1 viruses-15-02307-f001:**
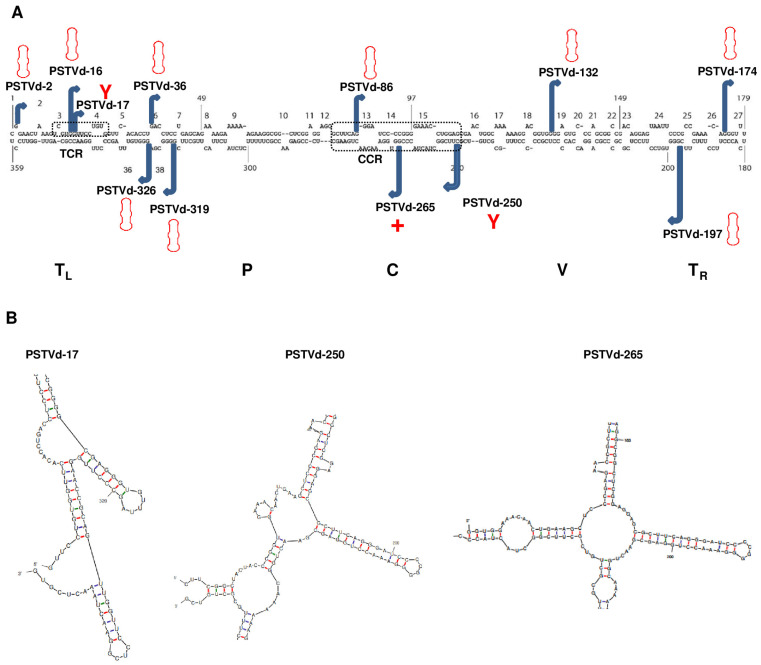
Secondary structure of the (+)-strand PSTVd genome and the diverse conformations of the 12 PSTVd forms studied. (**A**) The PSTVd genome consists of 359 nucleotides, forming a secondary structure comprising 26 stems and 27 loops. The PSTVd intermediate form exhibits a rod-like secondary structure, featuring the five hallmark domains typical of *Pospiviroidae* family members: Terminal Left (TL), Pathogenicity (P), Central (C), Variable (V), and Terminal Right (TR). The Central Conserved Region (CCR) resides within the C domain and includes a UV-sensitive loop E motif. The TL domain contains a Terminal Conserved Region (TCR). The loops, numbered from 1 to 27, are clearly indicated. The 12 PSTVd forms employed in this study were denoted by their respective start sites. Additionally, the rod-like, ‘Y’ shape, and ‘+’ shape conformations were also clearly marked. (**B**) Structural characteristics of ‘Y’ shaped PSTVd-17 and PSTVd-250, along with the ‘+’ shaped PSTVd-265, were depicted.

**Figure 2 viruses-15-02307-f002:**
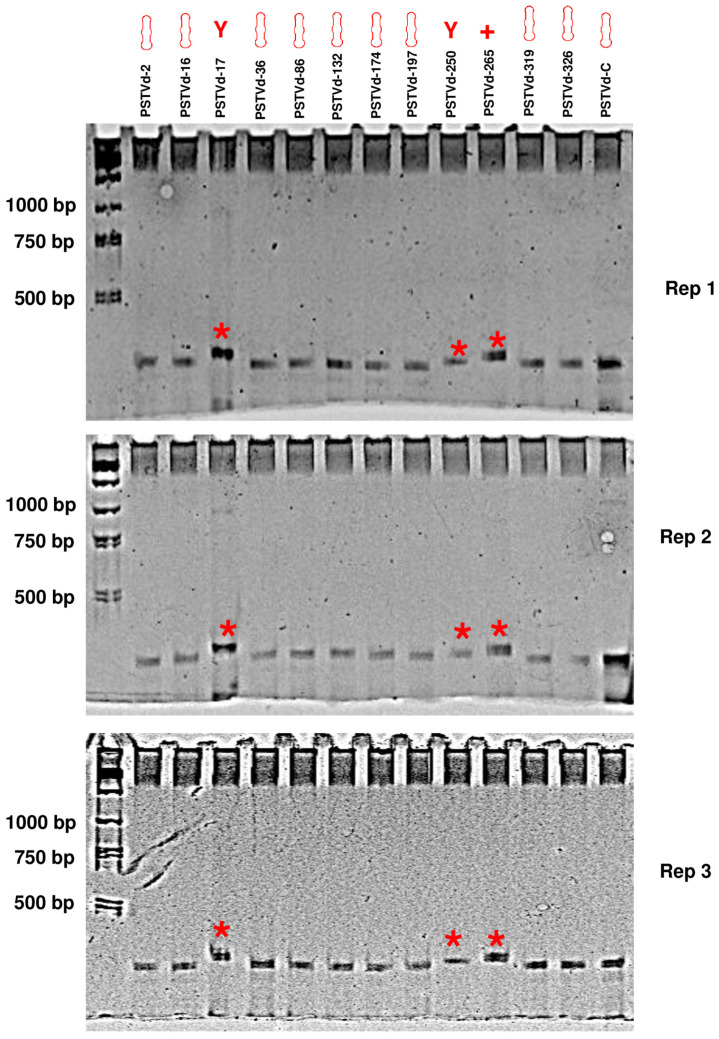
Confirmation of the secondary structure of the 12 PSTVd forms through native PAGE gel electrophoresis. Full-length PSTVd RNA transcripts for the 12 PSTVd forms were synthesized through in vitro transcription. Initially, these transcripts were purified using a 5% urea-PAGE gel, followed by separation through 5% native PAGE gel electrophoresis at 80 V for 4 h at 4 °C. Each sample contained 50 ng of RNA. As a reference, the circular PSTVd form (PSTVd-C), known for its rod-like structure, was included. Three biological replicates (reps) were incorporated, and images were captured after ethidium bromide staining. The three forms, PSTVd-17, -250, and -265, which form non-rod-like structures, were marked with a red asterisk.

**Figure 3 viruses-15-02307-f003:**
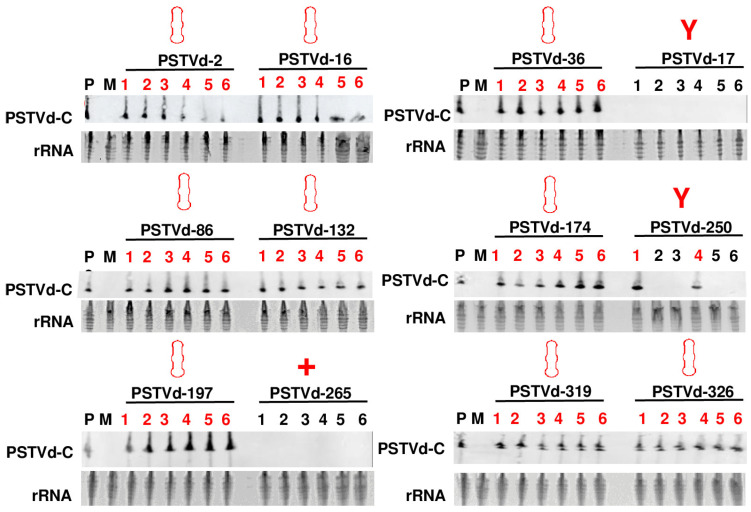
Analysis of the infection of 12 PSTVd forms in *N. benthamiana* plants. All 12 PSTVd forms underwent a plant infection assay using *N. benthamiana* plants, with six plants assigned to each form (numbers 1–6). Inoculation was performed through rub-inoculation, introducing 300 ng of PSTVd in vitro transcripts per plant. Systemic infection was assessed at 28 days post inoculation (dpi) using RNA blot. Gel electrophoresis was performed with 5% urea PAGE gel at 200 V for 4 h at room temperature. Each lane was loaded with 5000 ng of total RNA. As a reference, the circular PSTVd form (PSTVd-C), known for its rod-like structure, was included. Numbers in red indicated samples positive for PSTVd.

**Figure 4 viruses-15-02307-f004:**
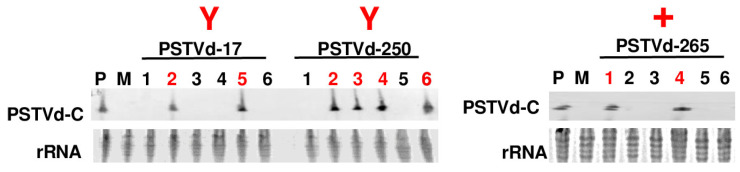
Infection of three PSTVd forms with non-rod-like structures following an increase in the quantity of linear RNA inoculum. The plant infection assay, as shown in Figure 3, was replicated for the PSTVd-17, PSTVd-250, and PSTVd-265 forms, utilizing an RNA inoculum of 3000 ng per plant. Position of circular PSTVd is marked by PSTVd-C. Systemic infection evaluation was conducted at 28 dpi. Six plants, numbered from 1 to 6, were included in each experiment. Numbers in red indicated samples positive for PSTVd.

**Table 1 viruses-15-02307-t001:** Primers used in RT-PCR to generate 12 PSTVd forms.

Forms	Forward (5′-3′)	Reverse (5′-3′)
PSTVd-2	TAATACGACTCACTATAGGAACTAAACTCGTGGTTCC	GAGGAACCAACTGCGGTTCC
PSTVd-16	TAATACGACTCACTATAGGTTCCTGTGGTTCACACC	ACGAGTTTAGTTCCGAGGAAC
PSTVd-17	TAATACGACTCACTATAGTTCCTGTGGTTCACACC	CACGAGTTTAGTTCCGAGGAAC
PSTVd-36	TAATACGACTCACTATAGACCTCCTGAGCAGAAAAG	AGGTGTGAACCACAGGAACC
PSTVd-86	TAATACGACTCACTATAGGGATCCCCGGGGAAACC	TGAAGCGCTCCTCCGAGCC
PSTVd-132	TAATACGACTCACTATAGGGAGTGCCCAGCGGCCGAC	CACCGTCCTTTTTTGCCAGTTC
PSTVd-174	TAATACGACTCACTATAGGGTTTTCACCCTTCCTTTCTTC	TGTTTCGGCGGGAATTACTCC
PSTVd-197	TAATACGACTCACTATAGGGTGTCCTTCCTCGCGC	GAAGAAAGGAAGGGTGAAAACCC
PSTVd-250	TAATACGACTCACTATAGCTTCGGCTACTACCCGGTG	CGACAGCGCAAAGGGGGC
PSTVd-265	TAATACGACTCACTATAGGTGGAAACAACTGAAGCTCC	GGGTAGTAGCCGAAGCGACA
PSTVd-319	TAATACGACTCACTATAGGGGCGAGGGTGTTTAGC	GAAGCAAGTAAGATAGAGAAAAAGC
PSTVd-326	TAATACGACTCACTATAGGGTGTTTAGCCCTTGGAACCG	TCGCCCCGAAGCAAGTAAGATAG

**Table 2 viruses-15-02307-t002:** Infection rates of 12 PSTVd forms.

Forms	300 ng per Plant	3000 ng per Plant
PSTVd-2	6/6	NA
PSTVd-16	6/6	NA
PSTVd-17	0/6	2/6
PSTVd-36	6/6	NA
PSTVd-86	6/6	NA
PSTVd-132	6/6	NA
PSTVd-174	6/6	NA
PSTVd-197	6/6	NA
PSTVd-250	2/6	4/6
PSTVd-265	0/6	2/6
PSTVd-319	6/6	NA
PSTVd-326	6/6	NA

Notes: NA, not available.

## Data Availability

All data supporting reported results can be found in the published paper.

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
