# Peer review of "The Secondary Structure of Potato Spindle Tuber Viroid Determines Its Infectivity in Nicotiana benthamiana"

_viruses, 2023, doi:10.3390/v15122307_

Round 1

Reviewer 1 Report

Comments and Suggestions for Authors

I understood that this paper discussed the secondary structure of viroids and their infectivity in tobacco. It is interesting to note the dependence of infectivity on secondary structure.

Author Response

We appreciate the reviewer's valuable comments.

Reviewer 2 Report

Comments and Suggestions for Authors

 Several questions arise about the interpretation of the results.

1. It is assumed that the DNA prepared by PCR has been transcribed to a  series of viroid RNAs with the same sequence but different termini and that the RNAs fold differently because they terminate at different points.  Can the authors indicate whether the termini occur across any of the reported functional regions of the circular native viroid?

2. Denaturing then non-denaturing gel electrophoresis has been used to prepare then analyse the structure of the RNAs. This shows that the predicted linear rods have the same mobility as the native circular viroid whereas the non-rodlike structures have lower mobility. However, the figures have been cropped at the bottom so that only one band can be seen, and the bands that are visible  are not sharp enough to show that they comprise a single or a double component.   Previous comparisons of the electrophoretic mobility of linear and circular forms of viroids show that they can be distinguished (eg. Nucleic Acids Research 25:4850-4854) and while the authors do not use 2-dimensional electrophoresis it is recommended that uncropped figures of the gels be shown to confirm that other components are not detectable using their staining method.

3. The infectivity assay used for comparing the linear viroid forms is not quantitative in that a single replicate of only 6 plants per treatment is statistically inadequate  to show significant differences in relative infectivity. Earlier studies used large numbers of tomato seedlings for quantitative assays. As no other relative infectivity assay was available for this viroid in this study  the authors rely on a single experiment to show that by increasing concentration of the non-linear form a larger number of plants are infected.  The same result might have been obtained if traces of an alternative form such as a circular form had been generated by ligation in vivo. It is recommended that replicated infectivity assays be included to support their conclusions about comparative infectivities.

Reviewer 3 Report

Comments and Suggestions for Authors

This article by Nie et al., is on “The secondary structure of Potato spindle tuber viroid determines its infectivity in Nicotiana benthamiana” is very interesting and written well. The author’s discussed about the Potato spindle tuber viroid (PSTVd), a non-coding infectious RNA, as a model to investigate the role of a specific rod-like structure in RNA function. They generated 12 PSTVd strains with different secondary structures and validated the folding structure with native PAGE-gel electrophoresis. Plant infection assay suggest the formation of rod-like structures and this might be needed for viroid replication and infection in N. benthamiana. However, certain questions are still unanswered. Material and methods section need to be improved more. Like which leaf the took for analysis systemic or other tissue? Some inoculated or symptomatic plant images need to be added (at least in supplementary). What the source and natural host of PSTVd? This article can be accepted after doing following changes;

Minor comments;

Page no. 1; Line no. 5-6; Authors affiliation should be numbered label.

Page no. 1; Line no. 17; mfold predication tool

Page no. 1; Line no. 21; RNA inoculum? Are you trying to say viral RNA transcript?

Page no. 2; Line no. 52; remove extra “.”

Page no. 2; Line no. 52; Mfold or mfold which one is correct?

Page no. 2; Line no. 77; what was the source of PSTVd intermediate strain? Is it the natural host?

Page no. 2; Line no. 84; What was the size of the amplified product?

Page no. 3; Line no. 110-111; Repeated lines, rewrite or delete this.

Page no. 4; Figure 1B; Image is not clear; can you provide a better image with a good resolution?

Page no. 4; Line no. 125; Check the heading format!

Page no. 4; Line no. 139; “reps”

Page no. 5; Figure 2; which molecular ladder did you used? Is it was having only 500bp? Labeling on the top of the gel need to be corrected for PSTVd-2 and replication 1, 2 and 3.

Page no. 5; Line no. 164; Not needed “Nicotiana benthamiana” Just write N. benthamiana

Page no. 5; Line no. 166; Why the authors have checked only one DPI? Which is 28 days? Why not early or late DPI?

Page no. 5; Line no. 169-170; I will suggest to make a tabular form of data of numbers of plants infected, no. of plants showed positive in testing etc. Do the infected plants show any symptoms? If yes, add some symptomatic plant image.

Page no. 6; Figure 3; 1-6 designates the number of plants? What is in the positive control? N. benthamiana infected plant or any other source plant? All the blot images should be labelled properly.

Page no. 6; Line no. 189; figurations? '+' shape

Page no. 7; Line no. 219; RNA infectious molecules

Page no. 7; Line no. 237; “effects [30]:

Page no. 7; Line no. 246; It has been noticed

Page no. 7; Line no. 247-248; reframe the sentence, meaning g is not clear.

All the blot images are too big, need to be reduce the size and labeled properly.

Comments on the Quality of English Language

Minor editing required. 

Reviewer 4 Report

Comments and Suggestions for Authors

The manuscript by Nie et al describes the proper folding of the PSTVd molecule is important for infectivity. This is an interesting manuscript based on a unique idea.

The following points need to be corrected.

Abstract

L15; I'm feeling uncomfortable using the term 'strain' in those like ‘12-PSTVd-strains’. I think the word 'strain' is used to mean an isolate with distinct properties, such as PSTVd severe strain. Since the molecules in this manuscript were prepared from different starting points of the same PSTVd template, I recommend using an appropriate word to indicate these. In L136, the expression 'PSTVd variants' can also be seen, but 'variants' is used in case such as 'sequence variants', which also does not seem to fit the molecule in this paper.

Introduction

L31; Watson-Crick (WC) base pairs are the most prevalent type of non-canonical pairing found in RNA.

Comment; Isn't Watson-Crick base pairing a canonical base pairing?

Results

L135-138; These strains were ----- 319, and -326.

Comment; A similar description can be seen on L84-86.

L147-150; RNA secondary structure prediction programs such as mfold will present multiple secondary structures. In the case of pospiviroids, the secondary structures predicted by mfold generally include a rod-shaped one and partly branched ones. Do the secondary structures shown in the results of this manuscript represent only those with the lowest free energy? In addition, was the secondary structure prediction made at 37°C?

L162 in footnote of Fig 1; PSTVd-2 is probably a mistake for PSTVd-17.

L204-213; The same results seem to be repeated for L204-208 "It was observed --- infected 0/6 plants" and L208-213 "At 28 days post inoculation --- lead to infection in any of the 6 plants". Please confirm.

Fig3. Please provide information on the electrophoresis conditions, such as gel concentration, PAGE or Agarose, denature or native, etc. in the footnotes, as Fig.2

Discussion

L269; “central domain (C)” can be changed to “C domain”.

L271l “central conserved region (CCR)” can be changed to “CCR”.

Comments on the Quality of English Language

no

Author Response

Dear reviewer,

Thank you for your valuable comments. We have thoroughly revised the manuscript in accordance with your suggestions. Kindly review the responses provided below for each question. The modified text was highlighted in red with a yellow shadow.

The manuscript by Nie et al describes the proper folding of the PSTVd molecule is important for

infectivity. This is an interesting manuscript based on a unique idea.

The following points need to be corrected.

Abstract

L15; I'm feeling uncomfortable using the term 'strain' in those like ‘12-PSTVd-strains’. I think the

word 'strain' is used to mean an isolate with distinct properties, such as PSTVd severe strain. Since the  molecules in this manuscript were prepared from different starting points of the same PSTVd template,  I recommend using an appropriate word to indicate these.

Reply-

In all places, strains have been replaced by forms.

In L136, the expression 'PSTVd variants' can also be seen, but 'variants' is used in case such as 'sequence variants', which also does not seem to fit the molecule in this paper.

Reply-

In all places, variants have been replaced by forms.

Introduction

L31; Watson-Crick (WC) base pairs are the most prevalent type of non-canonical pairing found in

RNA.

Comment; Isn't Watson-Crick base pairing a canonical base pairing?

Reply-

Sorry for the error. This sentence now reads- “Watson-Crick (WC) base pairs are the most prevalent type of canonical pairing found in RNA. ”

Results

L135-138; These strains were ----- 319, and -326.

Comment; A similar description can be seen on L84-86.

Reply-

Yes, however, the authors believe it is preferable to reiterate the description of the 12 forms in the Results section, recognizing that not all readers may read the Methods section.

L147-150; RNA secondary structure prediction programs such as mfold will present multiple

secondary structures. In the case of pospiviroids, the secondary structures predicted by mfold generally  include a rod-shaped one and partly branched ones. Do the secondary structures shown in the results of this manuscript represent only those with the lowest free energy? In addition, was the secondary structure prediction made at 37°C?

Reply-

Yes, we presented only the one with minimal free energy at 37°C.

Please check lines 77 and 78, we added the following sentence - “Only the one with the minimal free energy at 37°C was used.”

L162 in footnote of Fig 1; PSTVd-2 is probably a mistake for PSTVd-17.

Reply-

Sorry for this error. It has been corrected.

L204-213; The same results seem to be repeated for L204-208 "It was observed --- infected 0/6 plants" and L208-213 "At 28 days post inoculation --- lead to infection in any of the 6 plants". Please confirm.

Reply-

Sorry for this error. The repeated information has been deleted.

Fig3. Please provide information on the electrophoresis conditions, such as gel concentration, PAGE or Agarose, denature or native, etc. in the footnotes, as Fig.2

Reply-

Please check lines 222 and 223- “ Gel electrophoresis was performed with 5% urea PAGE gel at 200 V for 4 hours at room temperature. Each lane was loaded with 5,000 ng of total RNA. ”

DiscussionL269; “central domain (C)” can be changed to “C domain”.

Reply-

Revised.

L271l “central conserved region (CCR)” can be changed to “CCR”

Reply-

Revised.

Reviewer 5 Report

Comments and Suggestions for Authors

The authors generated 12 linear variants of PSTVd strain intermediate that differed by their starting position and tested these for infectivity. Three variants, which were predicted to have a bifurcating structure but not the standard rod-shape as minimum-free-energy structure, were less infectious than the others, for which rod-shaped minimum-free-energy structures were predicted. These three less infectious variants yielded some infections when the inoculum was increased from 300 ng/plant to 3 ug/plant. This fits to the biophysical knowledge on structure distributions of RNA. 

The manuscript does not give a reason on how or why the authors have selected the 12 presented linear PSTVd variants. 

Was there a difference on infectivity of the 9 variants, which where infectious with 300 ng/plant, when the inoculum was lowered to, f.e., 30 ng/plant?

The manuscript is a bit scarce on citations of general viroid topics and on publications with the same topic as this manuscript; for example 

https://doi.org/10.1002/j.1460-2075.1985.tb03914.x

https://doi.org/10.1016/0042-6822(92)90074-y

https://doi.org/10.1006/viro.1994.1491

https://doi.org/10.3390/cells10112971

Minor questions:

================

Line 158: "The TL domain can alternatively contain a Terminal Conserved Region (TCR)." The TCR region is just a sequence; how can the TL contain "alternatively" that sequence?

Lines 293: The given reference for alternative structural conformations and different functions of an RNA is fine, but you could also cite corresponding features in viroids: "native" versus self-splicing conformations with members of Avsunviroidae; involvement of the extra-stable hairpins, which are not part of the native structure, in replication and processing of Pospiviroidae members.

Lines 139--144 ("The secondary ... (TCR) [12--15]") are repeated in the legend of Fig. 1; thus delete them here.

Typos:

======

Lines 2, 13: Remove italics and uppercase from "Potato spindle tuber viroid"; see https://ictv.global/faq/names

Line 17: Delete "prediction"

Line 21: "viral" => "viroid"

Line 32: "non-canonical" => "canonical"

Line 36: "regions[6]" => "regions [6]" That is, add a space in front of references here and in further places.

Line 37: Do tRNAs and mRNAs exhibit catalytic functions?

Line 41: Ref. 9 might be not optimal; what about https://doi.org/10.1261%2Frna.049874.115 ?

Line 50: A species name has no abbreviation; see https://ictv.global/faq/names

Line 100: Leaves were "dusted" with Carborundum? Using a brush?

Line 121: "8 M urea" => "8 M urea followed by electrophoresis using ..."

Line 121: "SpeI"; Spe in italics

Lines 140, 155: "Pospiviroidae" in italics

Line 74, 148: Why are two different addresses given for mfold?

Line 174: "4h" => "4 h"

Lines 222, 246: "The functional form of PSTVd, known as circular PSTVd (PSTVd-C), was presented" => "Position of circular PSTVd is marked by PSTVd-C."

Author Response

We appreciate the reviewer's valuable comments. Please check our responses in the attached file.

Round 2

Reviewer 1 Report

Comments and Suggestions for Authors

accepted.

Author Response

(The authors gave the same response as above.)
